# Bottom-up synthesis of chiral covalent organic frameworks and their bound capillaries for chiral separation

Hai-Long Qian[1], Cheng-Xiong Yang[1] & Xiu-Ping Yan[1,2]

Covalent organic frameworks (COFs) are a novel class of porous materials, and offer great potential for various applications. However, the applications of COFs in chiral separation and chiral catalysis are largely underexplored due to the very limited chiral COFs available and their challenging synthesis. Here we show a bottom-up strategy to construct chiral COFs and an *in situ* growth approach to fabricate chiral COF-bound capillary columns for chiral gas chromatography. We incorporate the chiral centres into one of the organic ligands for the synthesis of the chiral COFs. We subsequently *in situ* prepare the COF-bound capillary columns. The prepared chiral COFs and their bound capillary columns give high resolution for the separation of enantiomers with excellent repeatability and reproducibility. The proposed strategy provides a promising platform for the synthesis of chiral COFs and their chiral separation application.

[1] College of Chemistry, Research Center for Analytical Sciences, State Key Laboratory of Medicinal Chemical Biology, Tianjin Key Laboratory of Molecular Recognition and Biosensing, Nankai University, 94 Weijin Road, Tianjin 300071, China. [2] Collaborative Innovation Center of Chemical Science and Engineering (Tianjin), 94 Weijin Road, Tianjin 300071, China. Correspondence and requests for materials should be addressed to X.-P.Y. (email: xpyan@nankai.edu.cn).

Covalent organic frameworks (COFs) are a novel type of crystalline porous materials with highly ordered structures solely constructed from organic building units via strong covalent bonds[1–5]. Owing to their strong covalent linkages between light elements (H, B, C, N and O), COFs possess lots of unique properties[4,5] such as rigid structures (two dimensional[1,6–8] or three dimensional[9,10]), low densities[9], high thermal stabilities[11] and permanent porosity with large specific surface areas[10], which make COFs potential in diverse fields including gas storage[12–14], photoconduction[15–17], catalysis[18–20] and chromatography[21].

Resolution of enantiomers is of great significant either in pharmacology or in biology, since pure enantiomers may profoundly differ in biological interactions, pharmacology and toxicity[22]. Chromatographic techniques based on chiral stationary phases have been proved to be one of the most attractive methods for separating and acquiring enantiopure compounds. Hence, exploring of novel porous materials such as metal organic frameworks[23,24], porous organic frameworks[25] and porous organic cages[26] as stationary phases in chiral chromatography has gained great concern recently. Considering the unique properties of COFs, the potential application of chiral COFs as stationary phases for chiral separation should be interesting and significant.

Although lots of COFs with multi-functions have been reported, the preparation of COFs with chiral functions has been rarely reported. Synthesis of a stable crystalline chiral COFs remains a major challenge that prevents their practical application in chiral catalysis and separation. The general strategies for introducing the chiral functional moieties into the COF networks can be summarized as post-synthesis and bottom-up strategies[27]. Recently, Jiang's group demonstrated the first example for the synthesis of chiral COFs via the post-synthesis approach[20]. S-pyrrolidine was utilized to functionalize the pore wall of a prepared COF via the subsequent coordinative incorporation to introduce the chiral centres and catalytically active sites in the COF frameworks. The synthesized chiral functionalized COF was then employed as a metal-free heterogeneous catalyst for chiral organocatalysts with high activity, enantioselectivity and recyclability.

Herein, we report a bottom-up strategy for the synthesis of chiral COFs and show an in situ growth approach for the fabrication of chiral COF-bound capillary columns for chiral separation. In this work, the 1,3,5-triformylphloroglucinol (Tp) is first functionalized with chiral (+)-diacetyl-L-tartaric anhydride ((+)-Ac-L-Ta) to form the chiral functionalized monomer CTp. 1,4-phenylenediamine (Pa-1), 2,5-dimethyl-p-phenylenediamine (Pa-2) and benzidine (BD) are then condensed with CTp to obtain chiral COFs CTpPa-1, CTpPa-2 and CTpBD, respectively. The prepared chiral COFs show high thermal stability. Furthermore, the fabricated chiral COF-bound capillary columns give high resolution for the separation of enantiomers. The results reveal the promising aspects for the fabrication of chiral COFs via the bottom-up strategy and the great potential of chiral COFs as a platform for chiral separation.

## Results

**Synthesis and characterization of chiral COFs.** Kandambeth et al. reported a series of COFs TpPa-1, TpPa-2 and TpBD with remarkable stability[28,29]. However, these COFs have no chiral function. Here we propose a bottom-up strategy to fabricate chiral COFs. Figure 1 illustrates the synthesis of chiral COFs. We chose the synthesis of CTpPa-1 as a proof of concept to show our bottom-up strategy. To construct the CTpPa-1, the chiral monomer CTp was first synthesized via the esterification of Tp and (+)-Ac-L-Ta (Fig. 1a; Supplementary Figs 1–3).

The obtained CTp and Pa-1 were then condensed in a proper solvent to form CTpPa-1 (Fig. 1b).

Solvent is a key factor to keep balance between framework formation and crystallization in the synthesis of highly crystalline COFs[5]. In our preliminary study, tetrahydrofuran (THF) was employed as the solvent to synthesize CTpPa-1 due to the good solubility of THF for CTp and Pa-1. However, only little red-coloured solid was obtained. In addition, the obtained solid was amorphous (Supplementary Fig. 4, green curve). To decrease the content of THF in the solvent, ethanol was added into the THF to form a binary solvent. The condensation reaction between CTp and Pa-1 was then conducted in a binary solvent containing ethanol and THF at the ratio of 1/1 (v/v). The resulting solid gave three peaks at 4.6°, 8.1° and 25.8° in its powder X-ray diffraction (PXRD) pattern, indicating the formation of crystalline structure (Supplementary Fig. 4, blue curve). Further decrease of the THF-to-ethanol ratio in the binary solvent led to the significant improvement of the diffraction peaks at 4.6°, suggesting the basic formation of better crystalline CTpPa-1 (Supplementary Fig. 4, red and black curves). However, THF is essential for the successful preparation of chiral CTpPa-1 as the diffraction peaks at 8.1° and 25.8° disappeared in the absence of THF (Supplementary Fig. 4, magenta curve). The good solubility of CTp in THF enables the high concentration of CTp, and thus leads to a rapid condensation and easy formation of an amorphous powder. Similar result was also found by Feng et al.[30] Therefore, a binary solvent of ethanol and THF (9/1, v/v) was selected for the preparation of chiral CTpPa-1.

Reaction time is another key factor for self-healing feedback process of reversible reaction to construct highly crystalline COFs. One hour reaction time gave no diffraction peaks (Supplementary Fig. 5, pink curve), and thus was insufficient to form crystalline structure. Increase of the reaction time to 2 h resulted in weak diffraction peaks at 4.6°, 8.1° and 25.8° (Supplementary Fig. 5, blue curve), suggesting that an ordered COF structure started forming. Increase of the reaction time to 4 h led to the significant improvement of all the diffraction peaks (Supplementary Fig. 5, red curve), indicating longer reaction time is benefit to the formation of better crystalline CTpPa-1. However, a reaction time of 6 h gave no further improvement of the diffraction peaks (Supplementary Fig. 5, black curve), suggesting that a reaction time of 4 h was sufficient for the formation of crystalline CTpPa-1.

The PXRD pattern of the as-prepared CTpPa-1 shows an intense peak at 4.6° and two weak peaks at 8.1° and 25.8° (Fig. 2a, red curve), suggesting that the prepared CTpPa-1 is a crystalline framework. The Material Studio (ver. 7.0) was used to simulate the optimum structure of the prepared CTpPa-1 (Supplementary Methods). Eclipsed AA and staggered AB models were generated and optimized to predict the structure of CTpPa-1 (Fig. 2d; Supplementary Fig. 6). The results imply that the CTpPa-1 mainly adopts the eclipsed AA stacking mode of a space group P6/m with $a = b = 21.9149$ Å, $c = 3.4824$ Å, $\alpha = \beta = 90°$ and $\gamma = 120°$ (Fig. 2d; Supplementary Table 1) as the PXRD experimental profile of CTpPa-1 (Fig. 2a, red curve) matches well with the simulated PXRD pattern of the eclipsed AA model after pawley refinement (Fig. 2a, blue curve) with the Rwp of 1.73% and the Rp of 1.35%, whereas the staggered AB model does not conform to the data (Fig. 2c; Supplementary Fig. 7).

The appearance of the Fourier transform-infrared spectroscopy (FT-IR) characteristic peak of CTpPa-1 at 1,664 cm$^{-1}$ (C=N) reveals the successful formation of imine bonds after the condensation of the aldehyde groups on CTp and the amino groups on Pa-1 (Fig. 3a, black curve). Meanwhile, the characteristic peaks of CTpPa-1 at 1,744 cm$^{-1}$ (C=O of ester) and 1,735 cm$^{-1}$ (C=O of carboxyl) imply the successful

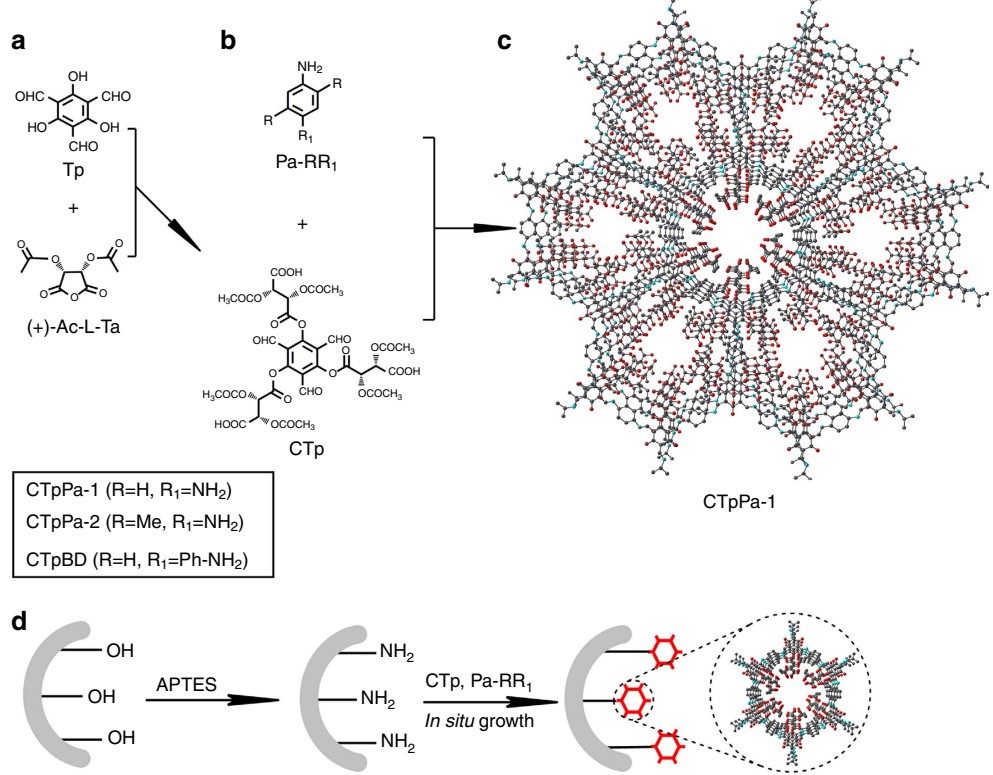

**Figure 1 | Synthesis of chiral COFs and their bound capillary columns.** (**a**) Synthesis of CTp through the esterification of Tp and ( + )-Ac-L-Ta. (**b**) Synthesis of chiral COFs through the condensation of CTp and Pa-RR1. (**c**) Graphic view of CTpPa-1 (C, grey; N, blue; O, red; H is omitted for clarity). (**d**) *In situ* synthesis of chiral COF-bound capillary columns.

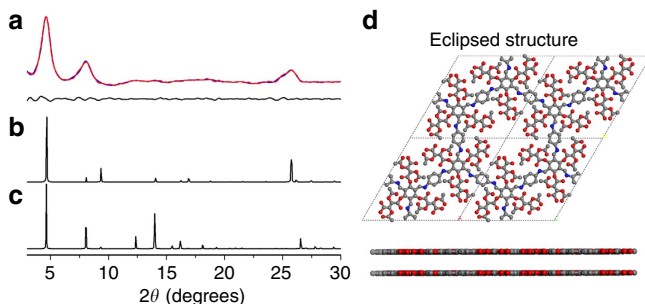

**Figure 2 | PXRD patterns and the structure of CTpPa-1.** (**a**) Experimental PXRD pattern of CTpPa-1 (red curve), refined modelling PXRD pattern of CTpPa-1 (blue curve) and the difference plot of the two PXRD patterns (black curve). (**b**) Simulated PXRD pattern of CTpPa-1 for the eclipsed model. (**c**) Simulated PXRD pattern of CTpPa-1 for the staggered model. (**d**) Eclipsed structure of CTpPa-1. C, grey; N, blue; O, red; H is omitted for clarity.

introduction of ( + )-Ac-L-Ta into the structure. In addition, the disappearance of N–H stretching peaks at $3,300–3,400\,cm^{-1}$ of Pa-1 (Fig. 3a, blue curve) as well as that of $C=O$ at $1,637\,cm^{-1}$ of the aldehyde on CTp (Fig. 3a, red curve) also confirms the formation of imine bonds via the condensation of CTp and Pa-1.

The structure of CTpPa-1 was further verified by solid-state $^{13}C$ nuclear magnetic resonance (NMR) spectroscopy (Supplementary Fig. 8). The $^{13}C$ NMR chemical shift of CTpPa-1 at 157 p.p.m. is ascribed to the carbon atom of the $C=N$ bond, which proves the formation of imine groups as well. The chemical shifts at 116, 120, 136, and 147 p.p.m. are assigned to the carbon atoms of phenyl groups, while chemical shifts at 21, 34, 40, 177

and 184 p.p.m. are attributed to the carbon atoms of chiral ligand groups.

Thermogravimetric analysis shows no obvious weight loss until up to 300 °C (Fig. 3b). In addition, the PXRD pattern of the CTpPa-1 remained unchanged after staying in air up to 250 °C for 1 h (Supplementary Fig. 9). The above results show the good thermal stability of the CTpPa-1. The scanning electron microscopy (SEM) and transmission electron microscopy images show that the CTpPa-1 is the fluffy aggregation of a sheet-like structure (Fig. 3c,d). The fluffy-layered sheet-like structure likely resulted from the strong π–π stacking interactions between adjacent layers, as confirmed from the simulated eclipsed AA structure of CTpPa-1.

The surface area and porosity of CTpPa-1 were measured by Argon adsorption–desorption analysis at 77 K (Fig. 3e,f; Supplementary Fig. 10). The Brunauer–Emmett–Teller (BET) surface area and the total pore volumes of the as-synthesized CTpPa-1 were 146 and $0.48\,cm^3\,g^{-1}$, respectively. The lower BET surface area and pore volume of CTpPa-1 likely resulted from the introduction of chiral groups that occupy the pore space and the less crystallinity in comparison with solvothermally synthesized TpPa-1 (ref. 28). The pore size of the CTpPa-1 calculated with nonlocal density functional theory was ~13 Å.

The proposed bottom-up methodology was also extended to prepare two more chiral COFs CTpPa-2 and CTpBD to better elucidate its versatility for the synthesis of chiral COFs (Supplementary Methods). CTpPa-2 and CTpBD were prepared with Pa-2 and BD instead of Pa-1 for CTpPa-1, respectively (Fig. 1), and characterized by PXRD, FT-IR, $^{13}C$ NMR spectroscopy, SEM, transmission electron microscopy and argon adsorption–desorption (Supplementary Figs 11–32; Supplementary Tables 2–3). The prepared CTpPa-2 and CTpBD show crystalline eclipsed AA structure with similar fluffy-layered sheet-like

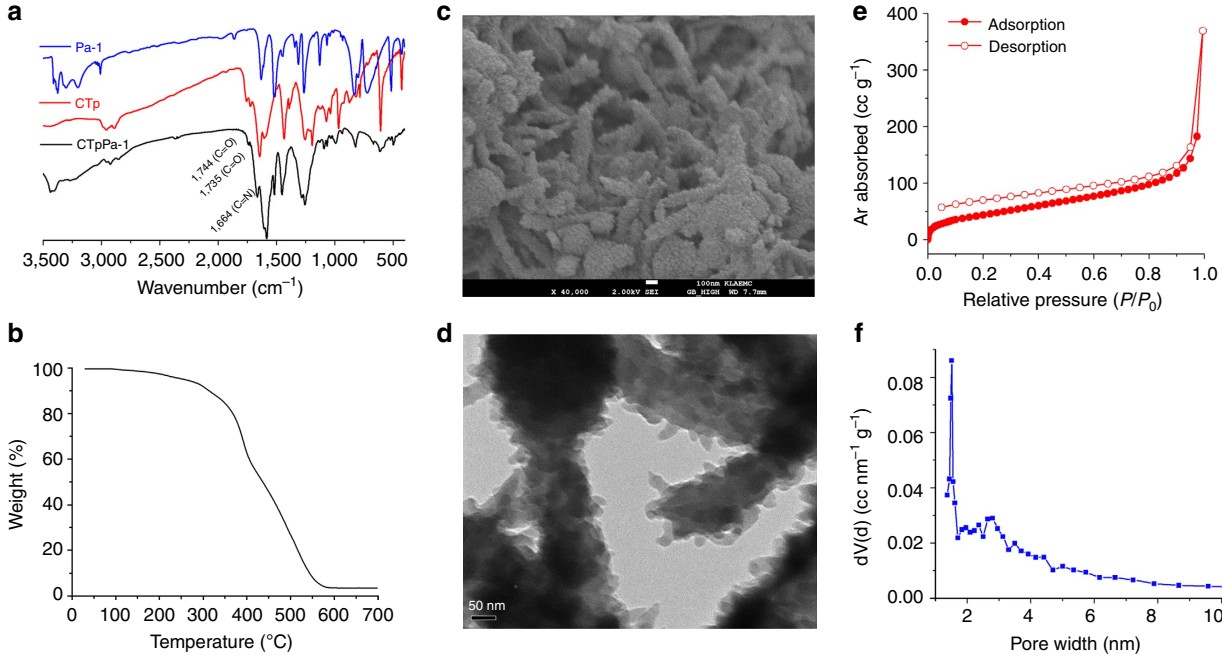

**Figure 3 | Characterization of CTpPa-1.** (**a**) FT-IR spectra of CTpPa-1, CTp and Pa-1. (**b**) Thermogravimetric curve of CTpPa-1. SEM (**c**) and transmission electron microscopy (**d**) images of CTpPa-1. (**e**) Argon adsorption–desorption isotherms of CTpPa-1. (**f**) Pore size distribution of CTpPa-1.

morphology to CTpPa-1 (Supplementary Figs 11–13,17,18,22–24, 28 and 29). However, CTpPa-2 gave smaller BET surface area $(104\,\mathrm{m^2\,g^{-1}})$ and pore size $(12\,\text{Å})$ than CTpPa-1 due to the additional methyl group of Pa-2 (Supplementary Figs 19–21), while CTpBD offered larger BET surface area $(317\,\mathrm{m^2\,g^{-1}})$ and pore size $(18\,\text{Å})$ than CTpPa-1 due to the extended phenyl of BD (Supplementary Figs 30–32).

**Fabrication and characterization of COF-bound capillaries.** The good thermal stability of the prepared crystalline chiral COFs encouraged us to explore their possibility for chiral separation. For this purpose, we used CTpPa-1 as a proof of concept to demonstrate the *in situ* growth approach for fabricating chiral COF-bound capillary column for chiral capillary gas chromatography (Fig. 1d). The fused-silica capillary column was first modified with 3-aminopropyltriethoxysilane (APTES) to provide the amino groups to react with the aldehyde groups on CTp. The solution of CTp and Pa-1 were then injected into the modified capillary column to *in situ* synthesize the CTpPa-1-bound capillary column.

The CTpPa-1-bound fused-silica plate was prepared in the same way as for CTpPa-1-bound capillary column to simulate the *in situ* growth of CTpPa-1 in capillary. The bound CTpPa-1 was then scraped from the fused-silica plate for PXRD characterization. The result shows that the PXRD pattern of the scraped CTpPa-1 is in agreement with the simulated one, revealing the successful *in situ* formation of CTpPa-1 on the surface of fused-silica (Fig. 4a). The appearance of the characteristic FT-IR peaks of the CTpPa-1 at $1,664\,\mathrm{cm^{-1}}$ (C=N), and $1,604\,\mathrm{cm^{-1}}$, $1,583\,\mathrm{cm^{-1}}$ and $1,520\,\mathrm{cm^{-1}}$ (phenyl groups) in the CTpPa-1-bound capillary column reveals the successful bonding of CTpPa-1 in capillary column (Fig. 4b). The SEM images (Fig. 4c,d cf. Supplementary Figs 33 and 34) and element mapping results (Fig. 4e,f cf. Supplementary Fig. 35) reveal a relatively uniform distribution of CTpPa-1 on the inner wall of CTpPa-1-bound capillary column. The CTpPa-2- and CTpBD-bound capillary columns were also prepared and characterized in a similar way

for CTpPa-1-bound capillary column (Supplementary Figs 37–41 and 44–48).

**COF-bound capillaries for chiral separation.** McReynolds constants are the typical parameters to evaluate the polarity of the stationary phase[31,32]. So, we measured the polarity of the chiral COF-bound capillary columns using benzene, 1-butanol, 2-pentanone, 1-nitropropane and pyridine as the probes (Table 1). The average McReynolds constants of chiral COF stationary phase range from 101.9 to 128.1, revealing a moderate polarity of the chiral COF-bound capillary columns. The chiral COF-based stationary phases show weak dispersion forces due to the low McReynolds constant for the X component (benzene). The maximum McReynolds constant for the Y component (butanol) among all the five test probes shows the strong hydrogen-bonding ability of the prepared chiral COF stationary phases. In addition, the McReynolds constants for the U (nitropropane), S (pyridine) and Z (2-pentanone) component indicate that the prepared stationary owns moderate electron donor and acceptor ability as well as part of dipolar and acidic character.

We then demonstrated the chiral resolution ability of the prepared chiral COF-bound capillary columns using capillary gas chromatography. We found that baseline separation of enantiomers such as ($\pm$)-1-phenylethanol, ($\pm$)-1-phenyl-1-propanol, ($\pm$)-limonene and ($\pm$)-methyl lactate on chiral COF-bound capillary columns within 5 min (Fig. 5; Supplementary Figs 42 and 49). However, ($\pm$)-1-phenyl-1-propanol and ($\pm$)-limonene were not separated on commercial β-DEX 225 and Cyclosil B chiral capillary columns, respectively (Supplementary Figs 52 and 53). Moreover, the chiral COF-bound capillary columns gave larger separation factors and better resolutions than commercial β-DEX 225 and Cyclosil B chiral capillary columns (Supplementary Table 4; Supplementary Figs 52 and 53). The above results show the good chiral separation performance of the fabricated chiral COF-bound capillary columns.

Even though the chiral recognition mechanism of enantioseparation is difficult to illustrate, the influence of the chiral

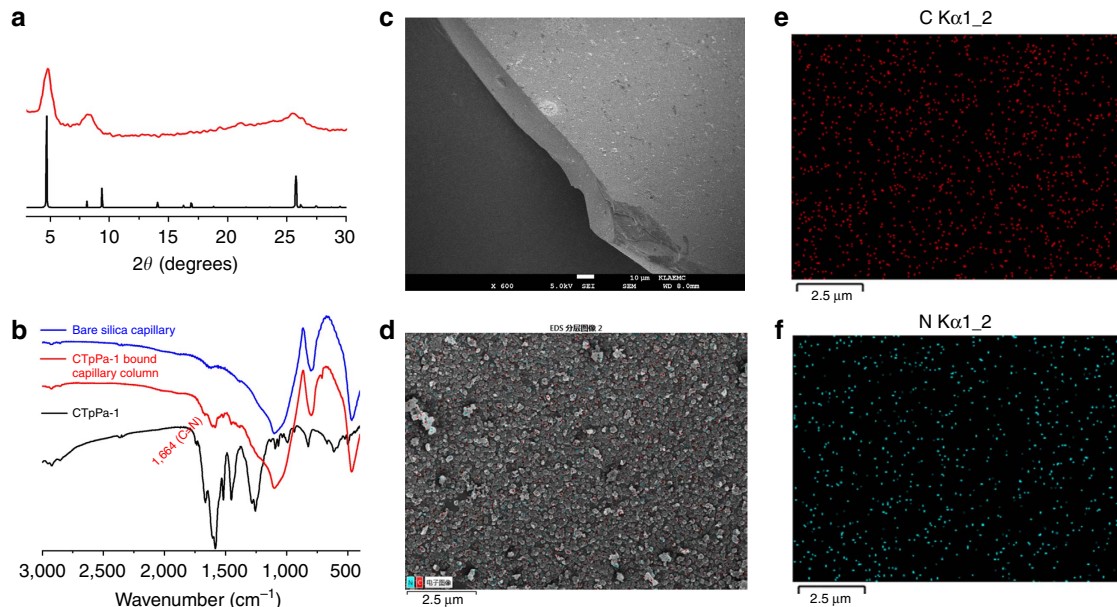

**Figure 4 | Characterization of CTpPa-1-bound capillary column. (a)** PXRD pattern of the CTpPa-1 scraped from the fused-silica plate (red curve) and the simulated PXRD pattern of CTpPa-1 (black curve). **(b)** FT-IR spectra of bare silica capillary column, CTpPa-1-bound capillary column and CTpPa-1. **(c)** SEM images of the edge of dissected CTpPa-1-bound capillary column. **(d)** Energy dispersive X-ray spectroscopy (EDS) element mapping images of the inner wall of CTpPa-1-bound capillary column. **(e,f)** C and N element mapping for the CTpPa-1-bound capillary column shown in **d**.

**Table 1 | McReynolds constants of chiral COF-bound capillary columns.**

| Column | X | Y | Z | U | S | Average |
|---|---|---|---|---|---|---|
| CTpPa-1 | 14.7 | 239.4 | 115.9 | 168.1 | 102.2 | 128.1 |
| CTpPa-2 | 16.5 | 227.3 | 105.8 | 149.2 | 90.1 | 117.8 |
| CTpBD | 24.9 | 189.1 | 94.7 | 115.4 | 85.6 | 101.9 |

COF, covalent organic framework.
Measured at 100 °C. X, Y, Z, U and S refer to benzene, butanol, 2-pentanone, nitropropane and pyridine, respectively.

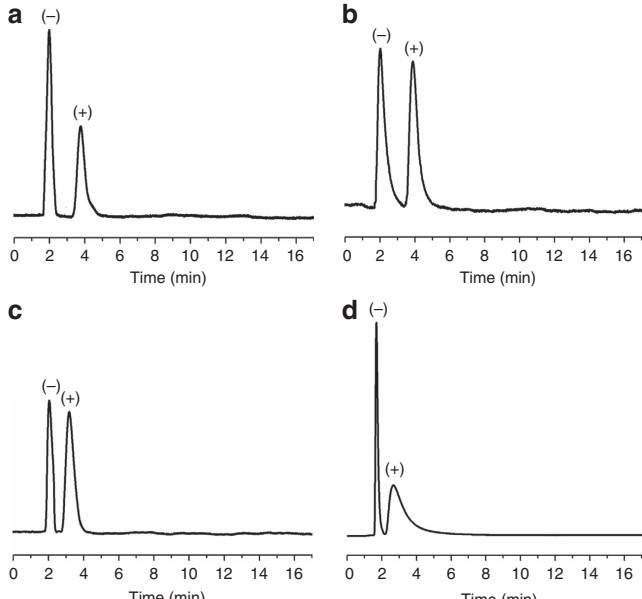

**Figure 5 | Gas chromatograms on CTpPa-1-bound capillary column (30 m long × 0.32 mm inner diameter). (a)** ( ± )-1-phenylethanol (200 °C, 1.5 ml min$^{-1}$ N$_2$); **(b)** ( ± )-1-phenyl-1-propanol (200 °C, 2 ml min$^{-1}$ N$_2$); **(c)** ( ± )-limonene (180 °C, 1.5 ml min$^{-1}$ N$_2$); **(d)** ( ± )-methyl lactate (170 °C, 1.5 ml min$^{-1}$ N$_2$).

microenvironment on the chiral properties of chromatographic systems is essential[33,34]. We also prepared ( + )-diacetyl-L-tartaric anhydride-functionalized capillary column (Supplementary Fig. 51) and found no chiral separation achieved on the column (Supplementary Fig. 54), indicating that only ( + )-diacetyl-L-tartaric anhydride cannot account for the good enantioseparation of the chiral COF-bound capillary columns. Other interactions such as van der Waals interaction, hydrogen-bonding and π–π interaction provided by the chiral COFs also affect the chiral chromatography resolution[35]. As the maximum kinetic diameters of the analytes 1-phenylethanol, 1-phenyl-1-propanol, limonene and methyl lactate (7.3, 7.3, 8.4 and 6.6 Å, respectively; Supplementary Fig. 55) are smaller than the pore size of the chiral COFs (12–18 Å), we assume that the chiral separation mainly occurred inside the pore of the chiral COFs. The combination of the charity of ( + )-Ac-L-Ta with the unique COF structures ensures the chiral microenvironment for chiral separation.

To further understand the retention and chiral discrimination of enantiomers, the enthalpy change (ΔH) and entropy change (ΔS), and the chiral part of enthalpy change (ΔΔH) and entropy change (ΔΔS) of the enantiomer–selector phase transfer were measured (Supplementary Methods). The determined thermodynamic parameters for the chiral separation of enantiomers on the chiral COF-bound capillary columns are summarized in Supplementary Tables 5–7; Supplementary Figs 36,43 and 50. All of the values of ΔH, ΔS, ΔΔH and ΔΔS are negative for the enantiomers studied, indicating both the retention and chiral

discrimination of the enantiomers on the chiral COF-bound capillary columns are driven by enthalpy. Moreover, all of the (+)-enantiomers have much larger negative entropy change than the (−)-enantiomers, indicating the (+)-enantiomers become more ordered in the microenvironment of chiral COFs. Meanwhile, such configuration fit is more favourable for the (+)-enantiomers to interact with the chiral COFs than the (−)-enantiomers, resulting in more negative values of $\Delta H$ for the (+)-enantiomers. The relatively smaller absolute values of $\Delta\Delta H$ for (±)-limonene and (±)-1-phenyl-1-propanol CTpPa-1- or CTpPa-2-bound capillary columns show the chiral discrimination of (±)-limonene and (±)-1-phenyl-1-propanol is due to steric hindrance with additional weak π–π interaction or/and hydrogen bond between (+)-enantiomers and the chiral COFs. In contrast, (±)-1-phenylethanol and (±)-methyl lactate give much larger absolute values of $\Delta\Delta H$, suggesting chiral discrimination seems to be the result of an additional strong π–π interaction or/and hydrogen bond for the most retarded (+)-enantiomers[36].

The repeatability and reproducibility of the chiral COF-bound capillary columns were investigated with CTpPa-1 as an example (Supplementary Table 8). The relative s.d. of the retention time for the run to run ($n = 7$), day to day ($n = 5$) and column to column ($n = 3$) were 0.15–0.30%, 1.11–1.89% and 2.35–3.41%, respectively, which demonstrates the excellent repeatability and reproducibility of the prepared chiral COF-bound capillary columns.

## Discussion

We have reported a bottom-up strategy to prepare chiral COFs, and an *in situ* growth approach to fabricate chiral COF-bound capillary columns for chiral gas chromatographic separation. The prepared chiral COFs CTpPa-1, CTpPa-2 and CTpBD exhibit two-dimensional eclipsed layered sheet structure with high thermal stability, while their bound capillary columns show high resolution for the separation of enantiomers with excellent repeatability and reproducibility. We find that the (+)-enantiomers show more ordered in the microenvironment of the chiral COFs with stronger interaction than the (−)-enantiomers, and that the high-resolution chiral separation is driven by enthalpy. The proposed strategy is easily to be extended to the preparation of other chiral COFs and their bound capillary columns. We believe that our work will promote the synthesis of chiral COFs and their wide applications in chiral separation and catalysis. Further attempt will be made to construct new chiral COFs by introducing the chiral functional groups on the diamine for catalysis or other applications.

## Methods

**Synthesis of chiral functionalized monomer CTp.** 1,3,5-Triformylphloroglucinol (Tp, 0.1 mmol) and (+)-diacetyl-L-tartaric anhydride (1.2 mmol) were separately dispersed in anhydrous THF (10 ml) under ultrasonication (80 W, 10 min). The Tp dispersion was dropwise added into the solution of (+)-diacetyl-L-tartaric anhydride at room temperature under Ar atmosphere. The mixture was refluxed at 60 °C for 24 h, then cooled down to room temperature. Water (10 ml) was added to the mixture to suspend the reaction. After the solvent was evaporated, the yellow-green residue (78.9 mg, 92% yield) was washed with water and dried under vacuum. $^1$H-NMR (DMSO-d6, 300 MHz, δ (p.p.m.)): 10.03 (s, 3H), 5.60 (d, 3H), 5.56 (d, 3H), 2.12 (s, 9H), 2.10 (s, 9H). FT-IR (powder): 3,451, 2,958, 1,756, 1,722, 1,644, 1,605, 1,433, 1,254, 1,071, 967, 607 cm$^{-1}$. MALDI-TOF-MS for $C_{33}H_{30}O_{27}$: 857.141 [M-H]$^−$ (calculated, 858.097).

**Synthesis of CTpPa-1.** Typically, a pre-reaction mixture containing CTp (0.1 mmol), Pa-1 (0.15 mmol), absolute ethanol (18 ml) and anhydrous THF (2 ml) was mixed under ultrasonication (80 W, 10 min) to obtain a homogeneous dispersion and then transferred into a three-necked flask equipped with a condenser. The mixture was refluxed at 80 °C for 4 h with Ar protection. The obtained dark-red-coloured precipitate was collected and washed with absolute THF and ethanol three times, then dried at 120 °C under vacuum for 24 h to get the

CTpPa-1 (77.5 mg) in the isolated yield of 76%. Analysis Calculated for $(C_{14}H_{12}NO_8)_n$: C 52.17; H 3.74; N 4.35. Found: C 50.92; H 3.88; N 4.15. FT-IR (powder): 1,744, 1,735, 1,664, 1,605, 1,583, 1,518, 1,452, 1,256, 822, 612 cm$^{-1}$. PXRD (2 theta): 4.6°, 8.1°, 25.8°.

**Preparation of CTpPa-1-bound capillary column.** A fused-silica capillary (30 m long × 0.32 mm inner diameter, Yongnian Optic Fiber Plant, Hebei, China) was treated sequentially with 1 M NaOH for 2 h, water for 30 min, 0.1 M HCl for 2 h, water again until the outflow reached pH 7.0, and methanol for 30 min. The pretreated capillary was filled with a methanolic solution of 3-aminopropyl-triethoxysilane (50%, v/v), and incubated in a 40 °C water bath overnight after both ends of the capillary were sealed with rubbers to obtain an amino-modified capillary. The amino-modified capillary was rinsed with methanol to flush out the residuals, and dried with a stream of nitrogen at 120 °C overnight for further use.

Solution A was obtained by suspending CTp (0.1 mmol) in ethanol (9 ml) and THF (1 ml), while solution B was obtained by dissolving Pa-1 (0.15 mmol) in ethanol (9 ml) and THF (1 ml). The above two solutions were cooled in an ice bath, and then mixed to obtain the pre-polymerization solution of CTpPa-1. The amino-modified capillary was quickly filled with the pre-polymerization solution with a syringe and incubated in an 80 °C water bath for 4 h after both ends of the capillary were sealed with rubbers. The prepared capillary column was rinsed with THF and ethanol to remove residuals, then flushed with N$_2$ for 2 h to remove the solvent. Finally, the prepared CTpPa-1-bound capillary column was conditioned with a temperature program: 80 °C for 30 °C min, ramp from 80 to 200 °C at a rate of 2 °C min$^{-1}$, and 200 °C for 1 h.

**Data availability.** The authors declare that the data supporting the findings of this study are available within the article and its Supplementary Information files.

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

## Acknowledgements

This work was supported by the National Basic Research Program of China (grant no. 2015CB932001), National Natural Science Foundation of China (grant nos 21435001 and 21305071) and Tianjin Natural Science Foundation (grant nos 14JCZDJC37600 and 14JCQNJC06600).

## Author contributions

H.-L.Q. acquired, analysed and interpreted the data, as well as developed the study concept and drafted the manuscript. C.-X.Y. instructed the synthesis experiments. X.-P.Y. conceived the study concept, guided the project, interpreted the data and wrote the manuscript.

## Additional information

**Competing financial interests:** The authors declare no competing financial interests.

