## [Peer review file · Nature Communications]

Reviewers' comments:

Reviewer #1 (Remarks to the Author):

This manuscript reported a bottom-up strategy to prepare chiral covalent organic framework (COF) for the first time, and the synthesized COF was then used as stationary phase of capillary gas chromatography column for chiral separation. The novelty includes:

- (1) A new bottom-up strategy to prepare chiral COF with good crystallinity and high thermal stability;
- (2) An in situ growth approach to fabricate the chiral COF-bound capillary column;
- (3) Successful application of above column for chiral separation, with high resolution and good reproducibility.

The authors also proposed the separation mechanism based on the well-designed experiment. In my view, the manuscript is properly organized, and the data are scientifically presented. All the conclusions are solidly supported by the data. Therefore, I recommend it to be accepted at its current form for publication in this journal.

Reviewer #2 (Remarks to the Author):

Comments

"Title: Bottom-up synthesis of chiral covalent organic framework and its bound capillary for chiral separation"

The authors present an interesting concept about introduction of chirality in a covalent organic framework and its application as stationary phase in bound capillary for chiral separation.

Major revision:

Comment 1: The same author research group has recently reported the fabrication of covalent organic framework to the capillary column. In that publication (Chem. Commun., 2015, 51, 12254) authors have shown high BET surface area as well thermal stability of the covalent organic framework, and separation of some selected alcohols in lesser time. Further, they have compared the separation performance of the TpBD coated capillary column with a commercially available HP-5 capillary column. Here the concept of synthesis of chiral covalent organic framework is not new but the synthesis using bottom-up strategy make it more interesting. However, this manuscript lacks comparison of chiral separation results of CTpPa coated capillary column with any commercially available chiral column.

Comment 2: The experimental results are not well presented including some characterization studies such as FTIR, Powder XRD, and SEM images need to be revised.

Comment 3: The authors have showed only one example of a chiral covalent organic framework (CTpPa). Some more examples should be included here at least two COFs as this would give better picture of methodology used here for the synthesis of chiral covalent organic framework. Thus, some additional experiments would need to be performed and few more covalent organic frameworks need to be added to the manuscript. Authors are advised to synthesize COFs with extended amine groups and compare the results and discuss.

Suggestion: Introduction of chiral functionality over 1,3,5-Triformylphloroglucinol (Tp) loses the keto-enol

tautomerism. This will lack the scope of resulting COF(s) to be used in acid/base catalyzed reactions. In this context, the authors are advised to introduce some chiral functional group(s) on diamine instead of on 1,3,5-Triformylphloroglucinol (Tp). This would extend the scope of resulting COFs for wider applications.

Comment 4: The authors have claimed, based on SEM and TEM images, that "CTpPa is the aggregation of numerous disk-like objects with lot of needles around the boundaries". However, from SEM images it appears to be some fluffy or amorphous compound aggregated there. In addition, it does not appear to have needle shape things around the boundaries. If this is not so, authors must provide more magnified SEM images in support of the statement.

Comment 5: The authors have observed "the Brunauer-Emmett-Teller (BET) surface area and the total pore volumes of the as-synthesized CTpPa were 146 m² g⁻¹ and 0.48 cm³ g⁻¹, respectively." The Brunauer-Emmett-Teller (BET) surface area and pore volume for the above mentioned CTpPa COF is very low. The authors should provide reason for this.

Comment 6: "The pore size of the CTpPa calculated with nonlocal density functional theory was about 1.3 nm". Furthermore, the authors have used this CTpPa-bound capillary column for chiral separation. The authors should discuss or provide some data that how this chiral covalent organic framework (CTpPa) having such small pore size exhibiting chiral separation and should compare the results with literature available. In addition to this, the authors should provide the mechanism for chiral separation happening here. And, is it occurring at surface or inside the pore?

Comment 7: The authors have simulated the in situ growth of CTpPa in capillary by preparing the CTpPa bound fused-silica plate in the same way. The bound CTpPa was then scraped from the fused silica plate for PXRD characterization. However, the Powder XRD pattern is not well matching with the simulated one. This shows the compound forming at CTpPa bound fused-silica plate is not the same one forming in situ inside the capillary. Furthermore, the FTIR spectral figures are not good and do not clarify the uniform distribution of CTpPa on the inner wall of capillary column.

Minor revision

Comment 8: In methods section the experimental Elemental Analysis data is not in well agreement with the calculated ones. Author is advised to revise the Elemental Analysis studies.

Comment 9: In the introduction paragraph correct the word "1,4-phenelynediamine" with "1,4-phenylenediamine."

Summary:

Overall, it is an interesting concept of introduction of chirality in covalent organic framework and their use for chiral separation. However, with less number of examples, poor characterization data, and no comparison of GC chiral separation results with available chiral GC column, this manuscript has lot of things missing and has no proper explanations at various places. Thus, this manuscript should not be considered for publication in Nature communication Journal.

Reviewer #3 (Remarks to the Author):

The work describes the synthesis of new chiral COFs for enantioselective separation in a GC column showing very good separation of a variety of substrates.

The paper describes in detail preparation of the COF, characterization via XRD, adsorption methods, SEM etc. but also the preparation of column and separation results. Moreover, evaluation of some thermodynamics is given to rationalize the mechanism.

Originality:

Certainly, this is not the first paper describing a chiral COF.

Also not the first paper describing enantioselective separation with a chiral COF. However, excellent progress in using GC columns could justify publication in Nat. Commun. The work is carried out quite carefully. And the topic is of interest for many groups.

Some minor criticism is the lack of detailed structural data due to the poor quality of XRD data. The authors should not overinterpret the results. Also the SEM data look more like the powder is not really forming a homogeneous coating. Why the performance is good for such an inhomogeneous coating? Can the authors compare the results to a molecular functionalized GC column with similar surface chemistry? Is it possible to get better insights into the structure of the coating with elemental mapping?

The clarity and writing is very good. Overall this could become a good contribution, certainly of interest to a wide readership.

POINT-BY-POINT RESPONSE TO REVIEWERS

The comments and suggestions made by the reviewers are very helpful for us to revise our manuscript. We highly appreciate the reviewers for such constructive comments. Detail reply to the comments and suggestions is made below.

Response to Reviewer 1

Comments:

This manuscript reported a bottom-up strategy to prepare chiral covalent organic framework (COF) for the first time, and the synthesized COF was then used as stationary phase of capillary gas chromatography column for chiral separation. The novelty includes:

- (1) A new bottom-up strategy to prepare chiral COF with good crystallinity and high thermal stability;
- (2) An in situ growth approach to fabricate the chiral COF-bound capillary column;
- (3) Successful application of above column for chiral separation, with high resolution and good reproducibility.

The authors also proposed the separation mechanism based on the well-designed experiment. In my view, the manuscript is properly organized, and the data are scientifically presented. All the conclusions are solidly supported by the data. Therefore, I recommend it to be accepted at its current form for publication in this journal.

Reply:

Thank you very much for your so positive comments.

Response to Reviewer 2

"Title: Bottom-up synthesis of chiral covalent organic framework and its bound capillary for chiral separation"

The authors present an interesting concept about introduction of chirality in a covalent organic framework and its application as stationary phase in bound capillary for chiral separation.

Comment 1:

The same author research group has recently reported the fabrication of covalent organic framework to the capillary column. In that publication (Chem. Commun., 2015, 51, 12254) authors have shown high BET surface area as well thermal stability of the covalent organic framework, and separation of some selected alcohols in lesser time. Further, they have compared the separation performance of the TpBD coated capillary column with a commercially available HP-5 capillary column. Here the concept of synthesis of chiral covalent organic framework is not new but the synthesis using bottom-up strategy make it more interesting. However, this manuscript lacks comparison of chiral separation results of CTpPa coated capillary column with any commercially available chiral column.

Reply and Corresponding Changes:

Thanks for your comments! In this work, we reports a novel bottom-up methodology for the preparation of new chiral COFs, and for the in situ fabrication of chemical bonding COF to inner wall of capillary for difficult and important chiral separation. In light of your valuable comments, we now make the comparison of our chiral COF bound capillary column with two commercial chiral capillary columns β -DEX 225 and Cyclosil-B. The results show that all of the enantiomers studied were baseline separated on our COF bond capillary columns, whereas some of them (+/-)-1-phenyl-1-propanol and (+/-)-limonene were not separated on commercial β -DEX 225 and Cyclosil B chiral capillary columns, respectively. Moreover, our chiral COF bound capillary columns gave much larger separation factors and better resolution than commercial β -DEX 225 and Cyclosil B chiral capillary columns (Please see the revised manuscript, page 11, line 5-11; revised Supplementary Information, Supplementary Table 4, Supplementary Fig. 52 and 53). In addition, the chiral COF bound capillary columns are extended to 30 m long to make a fair comparison with the 30 m long commercial capillary columns. Therefore, the corresponding data for 30 m CTpPa-1 bound capillary column are also updated (Please see the revised manuscript, Table 1; Fig. 5; revised Supplementary Information, Supplementary Fig. 36; Supplementary Table 5 and Table 8).

Comment 2:

The experimental results are not well presented including some characterization studies such as FTIR, Powder XRD, and SEM images need to be revised.

Reply and Corresponding Changes:

Good suggestions! In light of your suggestion, we now use improved SEM and TEM images to show the morphology of CTPa-1 (Please see the revised manuscript, Figure 3c and 3d). The FT-IR of the CTPa-1 bound capillary column is also improved (Please see the revised manuscript, Fig. 4b). The PXRD of CTPa-1 scraped from the fused silica plate is also revised, which matches with the simulated one now (Please see the revised manuscript, Fig. 4a). The SEM image of inner wall of CTPa-1 bound capillary column is now revised by photographing the dissected CTPa-1 bound capillary column (Please see the revised manuscript, Fig. 4c and 4d). The corresponding experimental results are now revised accordingly (Please see the revised manuscript, page 5, line 20; page 6, line 14; page 7, line 2; page 8, line 7; page 9, line 20-21).

Comment 3:

The authors have showed only one example of a chiral covalent organic framework (CTPa). Some more examples should be included here at least two COFs as this would give better picture of methodology used here for the synthesis of chiral covalent organic framework. Thus, some additional experiments would need to be performed and few more covalent organic frameworks need to be added to the manuscript. Authors are advised to synthesize COFs with extended amine groups and compare the results and discuss.

Reply and Corresponding Changes:

Thank you for your constructive suggestion. In view of your suggestion, we now add two more chiral COFs CTPa-2 and CTpBD to better elucidate the feasibility of our bottom-up strategy for the synthesis of chiral COFs as well as the in situ growth approach for the fabrication of chiral COFs bound capillary columns (Please see revised manuscript, page 8, the last paragraph to page 9, paragraph 1; revised Supplementary Information, page S3-S4; page S5, last paragraph to page S6, paragraph 1). The corresponding materials characterization data, and chromatographic data are also supplemented (Please see revised Supplementary information, Supplementary Fig. 11-32; Supplementary Fig. 37-50; Supplementary Table 2, 3, 6 and 7).

Comment 4:

Introduction of chiral functionality over 1,3,5-Triformylphloroglucinol (Tp) loses the keto-enol tautomerism. This will lack the scope of resulting COF(s) to be used in acid/base catalyzed reactions. In this context, the authors are advised to introduce some chiral functional group(s) on diamine instead of on 1,3,5-Triformylphloroglucinol (Tp). This would extend the scope of resulting COFs for wider applications.

Reply and Corresponding Changes:

It is a very useful advice to obtain chiral COFs for other more applications such as catalysis by introducing the chiral functional groups on diamine instead of on Tp. As the current work focuses on the chiral separation and the prepared chiral COFs offer good performance for such application, we will attempt your advice to construct new chiral COFs for catalysis or other applications in our future work.

Comment 5:

The authors have claimed, based on SEM and TEM images, that "CTpPa is the aggregation of numerous disk-like objects with lot of needles around the boundaries". However, from SEM images it appears to be some fluffy or amorphous compound aggregated there. In addition, it does not appear to have needle shape things around the boundaries. If this is not so, authors must provide more magnified SEM images in support of the statement.

Reply and Corresponding Changes:

Good suggestion. The SEM and TEM images for CTpPa-1 are updated in the revised manuscript (Please see revised manuscript, Fig. 3c and 3d). In addition, the descriptions about the morphology of prepared CTpPa-1 are now revised to "The scanning electron microscopy (SEM) and transmission electron microscopy (TEM) images show the CTpPa-1 is the fluffy aggregation of a sheet-like structure" (Please see revised manuscript, page 8, line 5-7).

Comment 6:

The authors have observed "the Brunauer-Emmett-Teller (BET) surface area and the total pore volumes of the as-synthesized CTpPa were 146 m² g⁻¹ and 0.48 cm³ g⁻¹, respectively." The Brunauer-Emmett-Teller (BET) surface area and pore volume for the above mentioned CTpPa COF is very low. The authors should provide reason for this.

Reply and Corresponding Changes:

Good question. The BET surface area and the total pore volume of porous materials were highly affected by the synthesis method likely due to the different crystallinity caused by different synthesis methods. For example, the BET surface area of TpPa-1 with good crystallinity was 535 m² g⁻¹ via the solvothermal synthesis method (J. Am. Chem. Soc. 2012, 134, 19524-19527), while the BET surface area of TpPa-1 with less crystallinity was 61 m² g⁻¹ through the mechanochemical synthesis method (J. Am. Chem. Soc. 2013, 135, 5328-5331). Moreover, the functionalized groups also gave significant effect on the surface area and the total pore volumes. For example, the BET surface area and total pore volumes of COF-LZU1 was 410 m² g⁻¹ and 0.54 cm³ g⁻¹. However, the BET surface area and total pore volumes of COF-LZU1 decreased to 146 m² g⁻¹ and 0.19 cm³ g⁻¹ after the functionalization of Pd (J. Am. Chem. Soc. 2011, 133, 19816-19822). Hence, The lower BET surface

area and pore volume of CTPa-1 likely resulted from the introduction of chiral groups and the less crystallinity in comparison to solvothermally synthesized TpPa-1 (Please see revised manuscript, page 8, line 13-16).

Comment 7:

"The pore size of the CTPa calculated with nonlocal density functional theory was about 1.3 nm". Furthermore, the authors have used this CTPa-bound capillary column for chiral separation. The authors should discuss or provide some data that how this chiral covalent organic framework (CTPa) having such small pore size exhibiting chiral separation and should compare the results with literature available. In addition to this, the authors should provide the mechanism for chiral separation happening here. And, is it occurring at surface or inside the pore?

Reply and Corresponding Changes:

Good question! The pore size of CTPa-1 (13 Å) is larger than that of the literature available MOFs {[ZnLBr]•H₂O}_n (1•H₂O) (12 Å) (Anal. Chem. 2014, 86, 1277-1281), [Cu₂(d-Cam)₂(4,4'-bpy)]_n (8.9 Å) (J. Chromatogr. A 2014, 1325, 163–170), [Zn₂(bdc)(l-lac)(dmf)]•DMF (5 Å) (J. Am. Chem. Soc. 2007, 129, 12958-12959) in chiral separation. In addition, the maximum kinetic diameters (calculated with ChemBio3D) of 1-phenylethanol, 1-phenyl-1-propanol, limonene, and methyl lactate are 7.3 Å, 7.3 Å, 8.4 Å, and 6.6 Å, respectively, which are all smaller than the pore size of the CTPa-1. Hence, the chiral separation mainly occurred inside the pore of CTPa-1.

Therefore, we now add the following explanation to address your question: "We also prepared (+)-diacetyl-L-tartaric anhydride functionalized capillary column (Supplementary Fig. 51) and found no chiral separation achieved on the column (Supplementary Fig. 54), indicating that only (+)-diacetyl-L-tartaric anhydride cannot account for the good enantioseparation of the chiral COF bound capillary columns. Other interactions such as van der Waals interaction, hydrogen-bonding and π-π interaction provided by the chiral COFs also affect the chiral chromatography resolution. As the maximum kinetic diameters of the analytes 1-phenylethanol, 1-phenyl-1-propanol, limonene and methyl lactate (7.3, 7.3, 8.4 and 6.6 Å, respectively; Supplementary Fig. 55) are smaller than the pore size of the chiral COFs (12-18 Å), we assume that the chiral separation mainly occurred inside the pore of the chiral COFs. The combination of the chirality of (+)-Ac-L-Ta with the unique COF structures ensures the chiral microenvironment for chiral separation" (Please see revised manuscript, page 11, last paragraph to page 12, paragraph 1).

Comment 8:

The authors have simulated the in situ growth of CTPa in capillary by preparing the CTPa bound fused-silica plate in the same way. The bound CTPa was then scraped from the fused silica plate for PXRD characterization. However, the Powder XRD pattern is not well matching with the simulated one. This shows the compound forming at CTPa bound fused-silica plate is not the same one forming in situ inside the

capillary. Furthermore, the FTIR spectral figures are not good and do not clarify the uniform distribution of CTpPa on the inner wall of capillary column.

Reply and Corresponding Changes:

Thank you for your suggestion! The PXRD of CTpPa-1 scraped from the fused silica plate is now re-experimented, which matches with the simulated one (Please see the revised manuscript, Fig. 4a). A clearer FTIR spectral is also provided in the revised manuscript (Please see the revised manuscript, Fig. 4b). In order to obtain the clear SEM image of the inner wall of the CTpPa-1 bound capillary column, we cut the column into two pieces, thus we can directly take the image of the inner wall (Please see the revised manuscript, Fig. 4c and 4d). Moreover, C and N element mapping of chiral COFs bound capillary column is supplemented to further exhibit the uniform distribution of chiral COFs on the inner wall of capillary column (Please see revised manuscript, page 10, paragraph 1; Fig. 4e and 4f; revised Supplementary information Fig. 33 - 35).

Comment 9:

In methods section the experimental Elemental Analysis data is not in well agreement with the calculated ones. Author is advised to revise the Elemental Analysis studies.

Reply and Corresponding Changes:

Thank you for your advice. The elemental analysis is now re-performed, and the data is now updated (Please see revised manuscript, page 14, line 18-19).

Comment 10:

In the introduction paragraph correct the word "1,4-phenelynediamine" with "1,4-phenylenediamine."

Reply and Corresponding Changes:

Thank you for your correction. We have corrected the word (Please see revised manuscript, page 4, line 13).

Response to Reviewer 3

Comment 1:

The work describes the synthesis of new chiral COFs for enantioselective separation in a GC column showing very good separation of a variety of substrates. The paper describes in detail preparation of the COF, characterization via XRD, adsorption methods, SEM etc. but also the preparation of column and separation results. Moreover, evaluation of some thermodynamics is given to rationalize the mechanism.

Originality:

Certainly, this is not the first paper describing a chiral COF.

Also not the first paper describing enantioselective separation with a chiral COF. However, excellent progress in using GC columns could justify publication in Nat. Commun. The work is carried out quite carefully. And the topic is of interest for many groups

Reply and Corresponding Changes:

Thank you very much for your positive comments. In view of your comments, “Considering the unique properties of COFs, the potential application of chiral COFs as stationary phases for chiral separation should be interesting and significant, however, has never been reported so far” is now revised to “Considering the unique properties of COFs, the potential application of chiral COFs as stationary phases for chiral separation should be interesting and significant” (Please see revise manuscript, page 3, line 16-18); “Herein, we report the first construction of chiral COF via a bottom-up strategy and also show the first fabrication of chiral COF-bound capillary column via an in situ growth approach for chiral separation” is now revised to “Herein, we report a novel bottom-up strategy for the synthesis of chiral COFs and show an in situ growth approach for the first fabrication of chiral COF bound capillary columns for chiral separation” (Please see revise manuscript, page 4, line 9-11).

Comment 2:

Some minor criticism is the lack of detailed structural data due to the poor quality of XRD data. The authors should not overinterpret the results. Also the SEM data look more like the powder is not really forming a homogeneous coating. Why the performance is good for such an inhomogeneous coating?

Reply and Corresponding Changes:

Good suggestion. In view of your suggestion, “..... corresponding to the diffractions from the (100), (200), and (001) facets, respectively” is now revised to “..... suggesting that the prepared CTpPa-1 is a crystalline framework (Please see the page 6, line 18-20); “The highly crystalline and good thermal stability of chiral COF CTpPa” is now revised to “The good thermal stability of the prepared crystalline chiral COFs” (Please see revise manuscript, page 9, line 8-10). “The results imply that the CTpPa adopts the eclipsed AA stacking

mode” is now revised to “The results imply that the CTpPa-1 mainly adopts the eclipsed AA stacking mode” (Please see revised manuscript, page 7, line 1-2).

In order to obtain the clear SEM image of the inner wall of capillary column, we cut the column into two pieces. The SEM data is updated with a dissected capillary column. The results show the homogeneous distribution of CTpPa-1 on the inner wall of the capillary column (Please see revised manuscript, Fig. 4c and 4d).

Comment 3:

Can the authors compare the results to a molecular functionalized GC column with similar surface chemistry?

Reply and Corresponding Changes:

Yes, thanks! In view of your comment, we also prepare the (+)-diacetyl-L-tartaric anhydride functionalized capillary column for comparison. We find that no chiral separation is achieved on the column, indicating that only (+)-diacetyl-L-tartaric anhydride cannot account for the good enantioseparation of the chiral COF bound capillary columns. The above information is now added (Please see revised manuscript, page 11, line 15-19; Revised Supplementary Information, Supplementary Fig. 51 and 54).

Comments 4:

Is it possible to get better insights into the structure of the coating with elemental mapping?

Reply and Corresponding Changes:

Yes! Thank you for your advice. The elemental mapping is now carried out. Element mapping results (Fig. 4e and 4f *cf.* Supplementary Fig. 35) reveal a relatively uniform distribution of CTpPa-1 on the inner wall of capillary column. The above information is now added (Please see revised manuscript, page 10, line 3-6; Fig. 4d, 4e and 4f; revised Supplementary Information, Supplementary Fig. 35).

REVIEWERS' COMMENTS:

Reviewer #2 (Remarks to the Author):

Authors have made a significant effort to address all the comments raised by the all the reviewers for their paper. However, as the reviewer 2 has mentioned the novelty of this paper still remains an issue as authors have already published a similar work on chromatographic separation in Chem. Commun., 2015, 51, 12254 and here they have only brought the chiral part in the paper.

Furthermore the COF(S) reported in this paper is constructed via C=N bonding. C=N bond formation is very much dependent on the bulkiness of the adjacent functionality. In this particular case authors have extremely bulky functionality adjacent to the -CHO functionality. This will surely prohibit the hexagonal COF formation as authors have predicted.

This is the reason the crystallinity is so poor among all the COFs reported in this paper. The principle reason is

that they have made a polymer with significant amorphous content, because the cyclisation has not happened. This is also a reason the porosity is quite low.

Authors also should do a refinement of the experimental PXRD and the theoretical one.

So I would recommend authors to make an attempt to put the chiral functionality on the amine not on the aldehyde, which in my opinion will result into a COF with much higher crystallinity and the surface area.

Given this problem regarding the characterisation of the main material, I am bit doubtful about recommending the acceptance. Hence I would let the editor to decide the final call on this aspect.

Reviewer #3 (Remarks to the Author):

This manuscript has been revised carefully and the referee comments were considered. Additional SEM data and mapping results confirm the homogeneity of the coating and the good results. This COF is certainly not as crystalline as a MOF and conclusions for the structure must be taken with care, but the data are presented in an appropriate way.

Unfortunately the pdf Format had some variations in the size of the figures, which made it difficult to read.

In summary the manuscript is suitable for publication in Nat. Commun.

POINT-BY-POINT RESPONSE TO REVIEWERS

Response to Reviewer 2

Comments:

Authors have made a significant effort to address all the comments raised by the all the reviewers for their paper. However, as the reviewer 2 has mentioned the novelty of this paper still remains an issue as authors have already published a similar work on chromatographic separation in Chem. Commun., 2015, 51, 12254 and here they have only brought the chiral part in the paper.

Furthermore the COF(S) reported in this paper is constructed via C=N bonding. C=N bond formation is very much dependent on the bulkiness of the adjacent functionality. In this particular case authors have extremely bulky functionality adjacent to the -CHO functionality. This will surely prohibit the hexagonal COF formation as authors have predicted.

This is the reason the crystallinity is so poor among all the COFs reported in this paper. The principle reason is that they have made a polymer with significant amorphous content, because the cyclisation has not happened. This is also a reason the porosity is quite low.

Authors also should do a refinement of the experimental PXRD and the theoretical one.

So I would recommend authors to make an attempt to put the chiral functionality on the amine not on the aldehyde, which in my opinion will result into a COF with much higher crystallinity and the surface area.

Given this problem regarding the characterisation of the main material, I am bit doubtful about recommending the acceptance. Hence I would let the editor to decide the final call on this aspect.

Reply:

Thank you very much for your comments. The novelty and significance of this work include the novel bottom-

up methodology for the preparation of new chiral COFs, and the new in situ fabrication of chemical bonding COF to inner wall of capillary, providing new possibilities for difficult and important chiral separation and catalysis. In addition, the high-resolution chiral separation reported on COFs in this work has never been achieved before. However, our previous work (Chem. Commun., 2015, 51, 12254) has nothing to do with new COFs, chiral COFs and chiral separation. So, our previous work (Chem. Commun., 2015, 51, 12254) has no effect on the novelty and significance of this work.

We greatly appreciate your suggestion to obtain chiral COFs for other more applications such as catalysis by introducing the chiral functional groups on diamine instead of on Tp. As the current work focuses on the chiral separation and the prepared chiral COFs offer good performance for such application, we will attempt your advice to construct new chiral COFs for catalysis or other applications in our future work (See revised manuscript, page 14, line 1-3).

However, we do not think the introduction of the chiral functionality on the amine on the aldehyde significantly affects the crystallinity and cyclisation because of very similar XRD patterns of the COFs with and without introducing the chiral functionality prepared under the same conditions. The good agreement between the experimental and theoretical XRD patterns (Figure 2) as well as the similar XRD patterns of the COFs with and without introducing the chiral functionality shows the occurrence of crystallisation and cyclisation. So, we think that the reduction of the porosity and surface area after introducing chiral functionality likely results from the occupation of the bulky chiral molecules in the pore space of the chiral COFs (See revised manuscript, page 8, line 13-16).

Actually, we did the refinement of the experimental PXRD and the theoretical patterns in Supplementary Information (Supplementary Tables 1-3; Fig. 2, Supplementary Fig. 11 and Supplementary Fig. 22), which you may not notice them.

Response to Reviewer 3

Comments:

This manuscript has been revised carefully and the referee comments were considered. Additional SEM data and mapping results confirm the homogeneity of the coating and the good results. This COF is certainly not as crystalline as a MOF and conclusions for the structure must be taken with care, but the data are presented in an appropriate way.

Unfortunately the pdf Format had some variations in the size of the figures, which made it difficult to read. In summary the manuscript is suitable for publication in Nat. Commun.

Reply:

Thank you very much for your positive comments. We have changed the figure size to make the figures more readable.